# Novel Biomarkers of Hepatitis B and Hepatocellular Carcinoma: Clinical Significance of HBcrAg and M2BPGi

**DOI:** 10.3390/ijms21030949

**Published:** 2020-01-31

**Authors:** Ian Baudi, Takako Inoue, Yasuhito Tanaka

**Affiliations:** 1Department of Virology and Liver Unit, Nagoya City University Graduate School of Medical Sciences, Nagoya 467-8601, Japan; baudiian@gmail.com; 2Department of Clinical Laboratory Medicine, Nagoya City University Hospital, Nagoya 467-8602, Japan; clinoue@med.nagoya-cu.ac.jp

**Keywords:** chronic hepatitis B (CHB), liver fibrosis, hepatocellular carcinoma (HCC), cirrhosis, intrahepatic covalently closed circular DNA (cccDNA), hepatitis B core-related antigen (HBcrAg), *Wisteria floribunda* agglutinin-positive Mac-2 binding protein

## Abstract

The hepatitis B virus (HBV) cannot be removed completely from infected hepatocytes, owing to the presence of intrahepatic covalently closed circular DNA (cccDNA). As chronic hepatitis B (CHB) can progress to cirrhosis and hepatocellular carcinoma (HCC), predicting HCC development in high-risk patients with high viral replicative activity or advanced fibrosis is important. Novel serological biomarkers reflect intrahepatic viral replicative activity or the progression of liver fibrosis, indicating non-invasive alternatives to liver biopsy: (1) Hepatitis B core-related antigen (HBcrAg) correlates with serum HBV DNA and intrahepatic cccDNA. In CHB patients, a decrease in HBcrAg is associated with favorable outcomes. HBcrAg can predict HCC occurrence or recurrence. (2) Measurement of the Mac-2 binding protein glycosylation isomer (M2BPGi) has been introduced for the evaluation of liver fibrosis. An increase in M2BPGi in CHB patients is related to the progression of liver fibrosis and high potential (risk) of HCC development. Here, we describe the clinical applications of HBcrAg and M2BPGi in CHB patients. Additionally, because new potential therapeutic agents that eliminate intrahepatic cccDNA are being developed, monitoring of HBcrAg or M2BPGi might be suitable for evaluating therapeutic effects and the clinical outcomes. In conclusion, these would be appropriate surrogate markers for predicting disease progression.

## 1. Introduction

Viral hepatitis is an infectious disease and a leading global killer [1]. The hepatitis B virus (HBV) causes acute and chronic infection, often leading to liver-related illness and accounting for over 600,000 liver-related deaths every year [2]. The majority of new HBV infections occur in highly endemic regions, such as China, Southeast Asia, and sub-Saharan Africa [3].

Chronic inflammation during active HBV infection is associated with continuing hepatocellular damage and tissue repair [4]. Consequently, liver fibrosis develops, with a progressive loss of liver function and increased risk of hepatocellular carcinoma (HCC). Up to 10% of chronic hepatitis B (CHB) patients may progress to severe fibrosis and cirrhosis, a major risk factor for HCC development [3].

Current HBV treatment primarily aims to prevent complications associated with progressive inflammation and fibrosis, i.e., liver failure, decompensated liver cirrhosis, and HCC [5,6]. Unfortunately, although nucleos(t)ide analogs (NAs) or interferons (IFNs) can effectively suppress HBV replication, these treatments are not curative [7]. This is because these drugs do not directly target the key molecule responsible for intrahepatic viral persistence, covalently closed circular DNA (cccDNA). cccDNA is a stable, extra-chromosomal transcriptional template for all HBV messenger RNAs (mRNAs) such as pregenomic RNA [8,9,10]. The amount and transcriptional activity of cccDNA in the hepatocytes are important for HBV disease progression and clinical outcomes [11]. 

To achieve effective clinical management of CHB, accurate laboratory data for diagnosis, treatment, monitoring, and prognostic assessment are essential (Figure 1). Liver biopsy, regarded as the gold standard for assessment of the degree of liver disease, is invasive and potentially risky [12,13]. Furthermore, it is prone to sampling errors and subjective result interpretation [14,15]. HCC diagnosis and surveillance are mostly based on the detection of tumor markers, such as alpha-fetoprotein (AFP) and protein induced by vitamin K absence or antagonist-II (PIVKA-II), and imaging techniques [5,6,16]. There remains the need for more reliable, non-invasive, and cost-effective biomarkers for CHB management. 

In this review, we evaluate two novel biomarkers showing great potential for HBV diagnosis and prognostic evaluation. The first is hepatitis B core-related antigen (HBcrAg), a surrogate marker of intrahepatic HBV replication that has shown good correlation with conventional HBV markers, such as HBV DNA and hepatitis B surface antigen (HBsAg) [17,18]. The second is Mac-2-binding protein glycosylation isomer (M2BPGi), which is a liver fibrosis marker with possibly predicting HCC development [19]. We focus on the clinical utility of these markers as predictors of HBV-related HCC development.

## 2. HBV Natural History and Biomarkers

### 2.1. HBV Replication Cycle

Hepatitis B virions (Dane particles) gain entry into the hepatocytes by binding to the receptor sodium taurocholate co-transporting polypeptide (NTCP) [20] and potential hepatocyte-specific co-receptors on the cell surface. The HBV envelope fuses with the membrane of the hepatocyte and the virion is endocytosed, releasing the viral DNA (partially double-stranded circular DNA), enclosed with the core particle, into the cytoplasm [21]. The viral envelope is lost (uncoating). The viral nucleocapsid containing the genomic DNA in a relaxed circular form is transported into the nucleus. In the nucleus, the viral DNA polymerase synthesizes fully double-stranded DNA, which is converted to cccDNA [21,22]. The formation of this cccDNA remains poorly understood but is most likely via a DNA repair mechanism [22] (repair and cccDNA formation). The cccDNA is then transcribed into the pregenomic and subgenomic mRNAs by the host RNA polymerase II [21,22] (transcription). Pregenomic RNA is the template for the translation of the DNA polymerase and core proteins, as well as being the template for reverse transcription. The DNA polymerase binds to the packaging signal of the pregenomic RNA and both are incorporated into immature core particles [21,22]. Reverse transcription of the pregenomic mRNA occurs in these immature core particles [21] (translation and reverse transcription). After synthesis of the (-) strand DNA, followed by the (+) strand DNA, the nucleocapsid, containing partially double-stranded circular DNA, matures (DNA synthesis). Early in infection, some mature nucleocapsids are recycled to the nucleus, building up a pool of around thirty cccDNA molecules in each infected hepatocyte. The mature nucleocapsids, containing partially double-stranded circular DNA, are enveloped by HBsAg to form complete virions (Dane particle) (assembly), which are then released from the infected hepatocytes [21,22] (release) (Figure 2).

The C gene encodes two major gene products, hepatitis B core antigen (HBcAg) (p21c), which forms the nucleocapsid, and a 22-kDa precore protein (p22cr), which is processed to hepatitis B envelope antigen (HBeAg, p17e) and secreted from the hepatocytes. Non-infectious particles (empty particles), composed of HBsAg, p22cr, and HBeAg, are also produced as a trap for the host immune system, to protect the infectious Dane particles [3].

### 2.2. Screening of Serological Markers Related to HBV Infection

The serological HBV biomarkers are described below: Serum HBV DNA; immunoglobulin M antibody to hepatitis B core antigen (IgM anti-HBc); immunoglobulin G antibody to hepatitis B core antigen (IgG anti-HBc); HBeAg and its corresponding antibody, anti-HBe; HBsAg and its corresponding antibody, anti-HBs. HBV RNA is also gaining attention as a possible marker of viral replication [23]. Serologic testing can diagnose acute or chronic HBV infection. In acute HBV infection, the first detectable serological markers are HBsAg and IgM anti-HBc [3].

### 2.3. Monitoring of Serological Markers Related to HBV Infection

The development of precise biomarkers can improve our understanding of the natural course of HBV infection and the optimization of anti-HBV treatments. Furthermore, because many CHB patients treated with antiviral therapy have undetectable HBV DNA, more precise biomarkers are required for HCC prediction. However, the ideal approach for anti-HBV treatment remains an unresolved problem. With the development of effective anti-HBV treatments, the prevention of disease progression and decrease in HCC occurrence in patients with CHB have been achieved [24].

Recently, because molecular investigations have been developed, several biomarkers related to the natural history of chronic HBV infection and usefulness of anti-HBV therapy have been recognized [24,25]. Conventional serological biomarkers including serum HBV DNA and HBsAg level can predict the risk of cirrhosis and HCC [26,27]. Quantitation of HBsAg has been used as a predictive marker of liver disease, spontaneous HBsAg seroclearance, cirrhosis, and HCC development, complementary to the measurement of HBV DNA [28,29,30,31]. However, even in CHB patients with undetectable HBV DNA and HBsAg seroclearance, cirrhosis and HCC can occur [32,33,34]. In addition, novel and useful biomarkers are still necessary to forecast treatment-induced or spontaneous HBeAg seroconversion, vital information, such as persistent responses before and after the cessation of NA treatment, and potential HBV reactivation and HBV reinfection after liver transplantation. We shall introduce HBcrAg as one of the best markers in the next section.

Of late, as other HBV markers, high-sensitivity HBsAg assay (HBsAg-HQ) and ultra-highly sensitive HBsAg assay as a semi-automated immune complex transfer chemiluminescence enzyme immunoassay (ICT-CLEIA) were reported by our group [35,36]. Due to the high sensitivity of ICT-CLEIA similar to that of HBV DNA quantification, ICT-CLEIA can be applied for HBV monitoring to prevent HBV reactivation-induced hepatitis [37].

## 3. HBcrAg

We have recently described the clinical application of a new effective biomarker, HBcrAg, which included the relationship between HBcrAg and other biomarkers, its capability for the prediction of clinical outcome, and its role as a treatment predictor [11]. Here, we introduce the distinctive features of HBcrAg and focus on its application as a predictor of HCC occurrence and recurrence.

### 3.1. Configuration of HBcrAg

HBcrAg consists of three products of the precore/core gene. HBcAg is a component of the virion and forms the nucleocapsid surrounding the viral DNA. p22cr is a 22-kDa precore protein present in HBV DNA-negative empty Dane-like particles [38]. HBeAg is a circulating peptide, derived from the precore protein by proteolysis and secreted from the hepatocytes [8,39]. All three proteins share an identical 149 amino-acid sequence [38,40]. HBcAg and HBeAg are the targets of cytotoxic T-cells and can induce host cellular immune responses [41]. By serological testing, HBcAg, p22cr, and HBeAg can all be measured as HBcrAg [42,43].

### 3.2. History of HBcrAg Testing and Development of Quantitative Assays

HBcrAg is now recommended in several clinical guidelines for CHB management, first in Japan, followed by the greater Asian region and, lately, in Europe [5,44,45].

In 2002, HBcrAg was first reported as a sensitive enzyme immunoassay specific for HBcAg and HBeAg [46]. Kimura et al. selected the precore/core gene products as HBcrAg, which comprised HBcAg and HBeAg. Serum samples for HBcrAg testing were treated to inactivate antibodies and to denature the antigen in advance, before the assay. This assay can detect HBcAg and HBeAg, even in specimens positive for anti-HBc or anti-HBe. HBeAg/anti-HBe seroconversion panels reveal that serum HBcrAg levels are paralleled by HBV DNA levels. As HBcrAg concentrations reflect the serum HBV DNA level, the test can be used supplementary to HBV DNA for the monitoring of CHB patients [46].

Further progress was achieved with the development of a CLEIA for the detection of HBcrAg. The clinical performance of this assay was evaluated in CHB patients, and the HBcrAg concentration was found to correlate positively with the HBV DNA concentration (p < 0.001) over a 100,000-fold range [42]. The accuracy of HBV load measurement obtained by the HBcrAg assay was not affected by the absence of HBeAg in the sera or the presence of precore mutations in the HBV genome. More complete information on the latest applications of HBcrAg measurement was provided in a recent review by Mak et al. [47]. Current quantitative HBcrAg assays have a lower limit of detection of 100 U/mL [37], but the current recommended cut-off line is 1000 (3 log) U/mL. Compared to “prototype HBcrAg assays” developed about 15 years ago, which measured only HBc, p22cr (empty particle), and HBeAg [47], more recently developed systems are automatic, as well as much more sensitive and speedy, by deactivating anti-HBc and anti-HBe in the test samples and using highly sensitive chemiluminescence-based detection systems [11].

### 3.3. Recent Clinical Assessments of HBcrAg

The novel biomarker, HBcrAg, has been used to support CHB monitoring and the prediction of clinical outcome. In this section, we briefly describe the clinical applications of HBcrAg, which have been reported recently from all over the world. Clinical applications of HBcrAg in CHB patients are summarized in Table 1.

Serum HBcrAg levels are closely associated with intrahepatic cccDNA levels, as well as serum HBV DNA [44,55]. Therefore, serum HBcrAg levels correlate with HBV markers of replication [44]. From France, Testoni et al. recently confirmed that serum HBcrAg levels in HBeAg-positive patients were significantly higher than those in HBeAg-negative patients without antiviral treatment. They also described that serum HBcrAg correlated with serum HBV DNA, intrahepatic pgRNA, and cccDNA levels, as well as transcriptional activity. In CHB patients, negativity for HBcrAg (<3 log U/mL) meant lower amounts of intrahepatic cccDNA and lower cccDNA activity compared to the patients positive for HBcrAg [44]. Riveiro-Barciela et al. in Spain reported that HBsAg levels <3 log IU/mL were only useful for identifying genotype D inactive carriers. A single HBcrAg measurement ≤3 log U/mL plus HBV DNA ≤ 2000 IU/mL was highly accurate in identifying inactive carriers, regardless of their HBV genotype [45].

In resource-limited countries or areas such as Africa, assays for HBV DNA quantification have limited availability and are expensive. Shimakawa et al. evaluated the potential of HBcrAg for identifying Gambian patients eligible for treatment, using a new experimental algorithm that does not include HBV DNA [56]. A total of 284 CHB patients without receiving antiviral treatment were analyzed. They examined the area under the receiver operating characteristic curve (AUROC), sensitivity, and specificity of serum HBcrAg referring the amount of serum HBV DNA. A simplified antiviral treatment algorithm using HBcrAg without HBV DNA presented a high AUROC (0.91 [95% CI: 0.88–0.95]), with a high sensitivity (96.6%) and specificity (85.8%) [56]. The measurement of HBcrAg, which is 5–10 times less expensive per test than the quantification of HBV DNA, has the possibility of replacing HBV DNA testing [56].

### 3.4. HBcrAg as a Predictor of HCC Occurrence

It is difficult to predict which patients will develop advanced liver disease, including HCC, during NA treatment [57]. High-HBV DNA titers have been correlated with an increased risk of liver cirrhosis and HCC development [27]. A low or undetectable HBV viral load reduces the risk of HCC development but does not inhibit it completely [54,58]. Several HBV markers have been shown to be related to HCC development in CHB patients, including the serum HBcrAg level [48,51,52]. 

In a large cohort study, HBcrAg was superior to HBV DNA in terms of predictive power for HCC development in treatment-naïve CHB patients [48]. During the follow-up period (median, 10.7 years), 78 of 1031 (7.6%) CHB patients without NA treatment experienced HCC development. Using HBcrAg levels, HBV DNA levels, HBeAg status, the covariates of HBV genotype status, and basal core promotor (BCP) status, a Cox proportional hazard model showed that HBcrAg > 2.9 log U/mL and BCP mutations were independently associated with HCC occurrence [48]. Meanwhile, Tseng et al. described that HBcrAg level is an independent risk factor of HCC in CHB patients with intermediate viral load (HBV DNA from 2000 to 19,999 IU/mL). The HBcrAg level of 4.0 log U/mL identifies patients with an intermediate viral load who are at high risk for HCC [49].

For treatment-experienced patients, NA reduced but could not eradicate the risk of HCC occurrence [58]. HBcrAg among 109 CHB patients receiving NA treatment for at least two years was an independent risk factor for HCC development (HR, 3.53) [51]. In 76 CHB patients receiving NA treatment with undetectable serum HBV DNA, the HBcrAg levels at baseline (pre-treatment) were also significantly higher in the patients with HCC development (5.45 log U/mL) than the matched control patients group (4.55 log U/mL) (p = 0.005). Additionally, a pre-treatment, HBcrAg > 4.67 log U/mL, predicted HCC incidence independently. Moreover, a post-treatment, HBcrAg > 3.89 log U/mL, predicted HCC with an odds ratio of 3.27. Considering only non-cirrhotic patients, a cut-off of >3.90 log U/mL predicted HCC with an odds ratio of 5.95 [50]. 

The continuing effect of NA treatment on HCC development was investigated by Kumada et al. CHB patients who received NA therapy were compared to those who did not [52]. In 57 of 234 patients (24.4%), HCC developed in the follow-up period. They demonstrated that greater age (HR 4.36 [95% CI, 1.33–14.29], *p* = 0.015), NA treatment (HR 0.28 [0.13–0.62], *p* = 0.002), BCP mutations (HR 12.74 [1.74–93.11], *p* = 0.012), high HBcrAg (HR 2.77 [1.07–7.17], *p* = 0.036), and high gamma glutamyl transpeptidase levels (HR 2.76 [1.49–5.12], *p* = 0.001) were factors significantly associated with HCC development. They concluded that higher HBcrAg levels and BCP mutations were associated with HCC development, independent of NA therapy [52].

To evaluate HCC occurrence in CHB patients, a combination of HBsAg and HBcrAg values was an effective biomarker [53]. In a recent study, 449 consecutive CHB patients were enrolled, and the association of HBsAg and HBcrAg with HCC occurrence was examined. If the cut-off values of HBsAg and HBcrAg were 3.0 and 3.0 log U/mL, respectively, CHB patients with HCC occurrence were often found in the low HBsAg (*p* = 0.002) and high HBcrAg (p < 0.001) groups. When HBsAg and HBcrAg were combined, an HCC occurrence was most frequent in the subgroup with low HBsAg and high HBcrAg, among the patients negative for HBeAg, regardless of NA therapy. In 338 patients without HCC occurrence at enrollment, HCC developed significantly more frequently in the low HBsAg/high HBcrAg group (p = 0.005) in a longitudinal analysis [53].

### 3.5. HBcrAg as a Predictor of HCC Recurrence

To predict HCC recurrence after resection or radio-frequency ablation, serological biomarkers have also been used. However, post-surgical HCC recurrence rates remained high regardless of NA treatment. In a recent report, HCC recurrence rates were up to 41.8% over two years [59]. In addition to tumor size, tumor number, macrovascular invasion, and cirrhosis, a pre-operative serum HBsAg titer >1000 IU/mL, baseline HBeAg positivity, and the prescription of NAs other than entecavir or tenofovir are associated with higher HCC recurrence [60,61]. 

A current study also confirmed that HBcrAg shows the predictive value for HCC recurrence after curative surgical treatment. In a study of 55 CHB patients receiving curative surgery for HCC, a serum HBcrAg level >4.8 log U/mL at the time of HCC diagnosis gave a high HR (8.96) for the following HCC recurrence within two years [54]. In another study, of 21 HCC patients who received liver transplantation, 5 patients (2 of 14 HBcrAg-positive patients and 3 of 7 HBcrAg-negative patients) developed HCC recurrence after liver transplantation. Conversely, the positivity of HBcrAg after liver transplantation was not correlated with HCC recurrence [62]. That is, the serum HBcrAg level before curative surgery might be a probable indicator to stratify post-surgical observation strategies and to find CHB patients with a high risk of HCC recurrence.

Moreover, in a recent report, the HCC recurrence-free survival rates were significantly lower in HCC patients with high intrahepatic cccDNA and serum HBcrAg levels than those with low cccDNA (p = 0.035) and low HBcrAg levels (P = 0.003) [63].

## 4. M2BPGi

Advanced glycoproteomic studies revealed that fibrosis leads to a specific modification of the glycosylation and sugar chain structure on the Mac-2 binding protein (M2BP) [64,65]. Based on these observations, M2BPGi has emerged as a novel surrogate serum glycoprotein-based biomarker (glycobiomarker) of the fibrosis stage. Here, we describe the development and latest evaluation of M2BPGi, not only as a surrogate marker of HBV-related liver fibrosis but also as a predictor of HBV-related HCC occurrence.

### 4.1. M2BPGi and Its Characteristics

Glycobiomarkers are novel biomarkers developed for the diagnosis of various diseases. In 2013, M2BPGi (also called *Wisteria floribunda* agglutinin-positive Mac-2 binding protein [WFA^+^-M2BP]) was introduced as a novel, noninvasive, rapidly assayed serological glycobiomarker for the evaluation of liver fibrosis [64]. 

M2BP has been found to have multibranching and sialylated N-glycans. WFA is thought to recognize the GalNAc residues of N-glycans and O-glycans and the clustered LacNAc (Gal-GlcNAc) structure [66]. M2BP is found in serum and in the extracellular matrices of many organs, including the liver [67]. As mentioned above, fibrosis results in specific modification of the glycosylation and sugar chain structure of M2BP, meaning that the levels of these modified M2BP proteins should correlate significantly with the progression of fibrosis [64,65].

Quantification assays of serum M2BP with fibrosis-specific modified sugar chain structures, i.e., M2BPGi, were developed, evaluated, and approved for clinical use in Japan [64,65,68]. M2BPGi is detected using a specific lectin called *Wisteria floribunda* agglutinin (WFA) that recognizes the N-acetylgalactosamine residue of N-glycans and O-glycans on M2BP [68], as depicted in Figure 3a. 

An automated and high-throughput assay can measure M2BPGi levels in 10 µL serum samples in 17 minutes [64]. Briefly, 10 μL serum is diluted and then mixed with WFA-coated magnetic bead particles. With magnetic separation and washing steps in between, ALP-labeled anti-M2BP monoclonal antibodies are then added, followed by chemiluminescent substrate reagents (Figure 3b). Light intensity is then measured and cut-off indices are calculated, based on assay control light intensity [64]. The HISCL M2BPGi assay (Sysmex Corp, Hyogo, Japan) has a reported range from 0.1 to 20 cutoff index (C.O.I.), and samples with less than 1.0 C.O.I. are considered negative.

### 4.2. M2BPGi as a Predictor of Hepatocarcinogenesis and Marker of Fibrosis in Chronic Hepatitis C (CHC)

In patients with CHC, factors such as older age, male gender, and liver fibrosis have been reported as predictors of hepatocarcinogenesis [69]. As the presence of severe liver fibrosis or cirrhosis prior to treatment is generally recognized as a higher risk factor for HCC development [70], even after a sustained virological response (SVR) is achieved [71,72], the assessment of the fibrosis stage is of great importance for CHC patients.

Diagnosis of cirrhosis is difficult if the laboratory data are confirmed as normal or near normal at the compensated stage [73]. Liver biopsy has long been considered to be the gold standard for the staging of liver fibrosis [74,75]. Nevertheless, liver biopsy has limitations because it is an invasive procedure with the risk of severe adverse events, and there may be sampling errors due to various states of fibrosis in the liver [76]. Therefore, a serum fibrosis biomarker such as M2BPGi should overcome these limitations and provide improved diagnostic accuracy.

M2BPGi was originally reported as a fibrosis-related glycol-alteration, and a significant relationship between the serum M2BPGi level and histological stage of fibrosis was confirmed [73]. Moreover, several demographic variables correlated with serum M2BPGi levels: Greater age, male gender, and higher body mass index (BMI), are associated with higher serum M2BPGi levels [73]. Remarkably, some of them have been reported as risk factors for HCC development [77]. Therefore, serum M2BPGi levels possibly reflect HCC risk factors other than liver fibrosis [73]. Additionally, the cumulative survival rate was significantly lower in cirrhosis patients with M2BPGi levels of ≥5.0 C.O.I. than in cirrhosis patients with M2BPGi levels <5.0 C.O.I. [78]. Sato et al. also reported that high serum M2BPGi is an independent risk factor for HCC development after achievement of SVR. A serum M2BPGi of ≥2.80 is associated with a >15-fold-increased risk of HCC development in CHC patients who have achieved SVR [73].

### 4.3. M2BPGi for HCC Prediction in CHB

M2BPGi has been used predominantly for evaluation of liver fibrosis in CHC patients. However, the diagnostic value of M2BPGi for assessing liver fibrosis has been demonstrated in patients with CHB, non-alcoholic fatty liver disease, and biliary atresia, as well as CHC [64,79,80,81,82]. In addition, M2BPGi levels are prognostic of the risk of HCC development with HBV or HCV infection [80,83]. In this section, we summarize that M2BPGi shows good performance for the assessment of liver fibrosis and HCC prediction in CHB.

Serum M2BPGi can predict the HCC development in CHB patients [83]. Kim et al. reported that the M2BPGi level was an independent predictor of HCC development (adjusted hazard ratio; 1.143, 95% CI; 1.139–1.829), along with male gender and diabetes (all *p* < 0.05) in multivariate analysis. In patients without cirrhosis (n = 1,087), M2BPGi levels ≥1.8 were associated with a higher risk of HCC development (*p* < 0.001 by log-rank test), whereas M2BPGi levels ≥1.8 tended to be associated with a higher risk of HCC development in patients with cirrhosis (n = 236) (*p* = 0.073 by log-rank test) [83]. In another report, Liu et al. described that M2BPGi was a strong and independent short-term predictor of HCC in CHB patients [84].

Moreover, to predict HCC occurrence, Jun et al. described that M2BPGi was more effective than AFP, and M2BPGi was an independent predictor of HCC [19]. Nine-hundred and forty-seven treatment-naïve patients who were mono-infected with HBV or HCV and without HCC at baseline were enrolled. The median M2BPGi was significantly higher among patients with cirrhosis (2.67 C.O.I.) than non-cirrhosis (0.80 C.O.I.) (*p* < 0.001) and those who developed HCC (3.22 C.O.I.) than those who did not (1.16 C.O.I.) (*p* < 0.001). Although the AUROC for M2BPGi and AFP was comparable overall (0.77 versus 0.72; *p* = 0.15), M2BPGi outperformed AFP for CHB patients (0.84 versus 0.75; *p* = 0.02). 

In a recent report, M2BPGi could be used to accurately identify CHB patients with a low risk of HCC. Mak et al. evaluated the value of M2BPGi for HCC prediction in 207 non-treated HBeAg-negative patients with recognized HBeAg seroconversion [85]. M2BPGi was measured at baseline, 5 years, and 10 years after seroconversion in this follow-up study (median; 13 years). Baseline M2BPGi precisely predicted not only HCC (OR; 4.66) but also advanced fibrosis at the 5 years point, as determined by liver stiffness measurement. Using a cut-off of 0.68 C.O.I, the baseline M2BPGi showed good performance (AUC, 0.88; sensitivity, 90%; and specificity, 80%). In these HBeAg-negative patients, the negative predictive value was very high (99.3%), while the positive predictive value was low (25.8%) [85].

In CHB patients receiving NA treatment, the M2BPGi level before treatment and 48 weeks after the start of NA treatment is an indicator of HCC occurrence. Hsu et al. mentioned that the baseline M2BPGi level can be a factor of the prediction of HCC occurrence in patients with cirrhosis treated with NAs. Moreover, in patients with cirrhosis, the baseline M2BPGi level was associated with HCC occurrence, while the M2BPGi levels at year 1 or 2 were not independently predictive. As a result, a risk score for HCC occurrence was developed using the baseline M2BPGi, age, and body mass index at 3, 5, and 10 years, respectively [86]. Shinkai et al. also reported that serum M2BPGi levels ≥1.215 C.O.I. at 48 weeks were associated with HCC development (HR 5.73; *p* ≤ 0.001) [87].

In patients with undetectable HBV DNA during NA therapy, a higher pre-treatment M2BPGi level is associated with an increased risk of HCC development [86,88]. Cheung et al. compared 57 NA-treated patients, with undetectable HBV DNA and HCC development, with 57 controls. There was a significant difference in the median levels of pre-treatment M2BPGi between the HCC and control groups (0.67 vs. 0.41 C.O.I., respectively, *p* < 0.001). Among patients with cirrhosis, the median level of M2BPGi was higher in the HCC group than the control group (0.74 vs. 0.47 C.O.I., respectively, *p* = 0.014). Among patients without cirrhosis, the median level of M2BPGi of the HCC group was also higher (0.48 vs. 0.28 C.O.I., respectively, *p* = 0.002). With a cut-off value of 0.69 C.O.I., the AUROC of pre-treatment M2BPGi to predict HCC development for the whole cohort was 0.70 C.O.I. With 0.69 and 0.34 C.O.I., the AUROCs to predict HCC were 0.67 and 0.77 C.O.I. for patients with and without cirrhosis, respectively [88].

Therefore, M2BPGi might be suitable for assessment of the level of fibrosis and prediction of HCC in CHB patients. Clinical applications of M2BPGi in patients with chronic liver disease are summarized in Table 2.

## 5. Discussion and Conclusions

In this review, we have described the characteristics and clinical applications of two novel biomarkers. HBcrAg is appropriate for evaluating the amount of intrahepatic cccDNA in CHB patients and M2BPGi can assess both the severity of liver fibrosis and the risk of HCC development in patients with CHB. Here, we discuss both HBcrAg and M2BPGi regarding the future prospects and progress required.

First, regarding HBcrAg, the serum HBcrAg level is associated with the serum HBV DNA in all CHB clinical stages. Even in patients achieving a “functional cure” with undetectable serum HBV DNA and HBsAg, severe difficulties including HBV reactivation, cirrhosis, and HCC occurrence are described. As some patients achieving a “functional cure” still show detectable serum HBcrAg, future studies comparing the long-term outcome between HBcrAg-positive and HBcrAg-negative patients are essential to improve the outcome of CHB patients. The clinical practice based on HBcrAg needs additional investigation and argument. Recently, the number of reports on the topic of HBcrAg has been increasing. To date, most reports have been published from East Asia. For HBcrAg to be used more in clinical practice, large cohort studies should be completed in other areas, especially in the US and Europe.

Regarding M2BPGi, this has been used as a valuable biomarker to evaluate liver fibrosis, especially in CHC patients. Based on its background, the majority of reports on the topic of M2BPGi are about CHC, and there are fewer reports about CHB [79,89,90]. That is, although past efforts have documented the accuracy of M2BPGi assay in patients with CHC, few have addressed CHB. 

The M2BPGi cut-off value for predicting cirrhosis or HCC varies, depending on the etiology. Ichikawa et al. reported that M2BPGi ≥ 0.71 is one of the risk factors for HCC development in CHB patients [91]. Nishikawa et al. categorized the levels of M2BPGi, based on the fibrosis stage diagnosed by transient elastography. The median levels of M2BPGi for F2, F3, and F4 were 1.49, 1.79, and 2.83 C.O.I., and 3.19, 3.79, and 5.03 C.O.I., in CHB and CHC, respectively [90]. As described above, M2BPGi levels ≥1.215 after 48 weeks of starting NA treatment [87] were associated with HCC. Considering the results of these reports, the M2BPGi levels of CHB patients seem to be lower than those of CHC patients. Conversely, in CHC, independent of liver fibrosis, whether the M2BPGi level is increased is still a matter of debate. Additionally, there is only one report evaluating whether M2BPGi can predict HCC occurrence in non-Asian patients and patients outside of East Asia [19]. Further studies are required to assess serum M2BPGi level as an HCC biomarker in extensive patient populations, including non-Asian patients, especially those with severe liver fibrosis.

Anti-HBV treatment to eliminate cccDNA from hepatocytes is important for suppressing HCC occurrence. Although effective anti-HBV treatments, such as IFN-based regimens and NAs, have been advanced, they are not perfect, because cccDNA resides in the hepatocytes. As a new class of potential therapeutic agents, capsid assembly modifiers can directly inhibit the intrahepatic core protein and possibly cccDNA as well [92]. Additionally, because all cccDNA-derived HBV transcripts overlap at their 3’ ends, the short interfering RNAs in ARC-520 were designed to target all HBV mRNAs and reduce antigenemia, as well as enable potential host immune responsiveness and a functional cure [93]. 

To confirm a functional cure of HBV, higher-sensitivity assays for both HBsAg and HBcrAg are required. At present, new HBcrAg assays with approximately 10-fold higher sensitivity are being developed (Tanaka Y, The Liver Week 2019 in Busan). In addition, because a fully automated pretreatment technique before HBcrAg measurement is convenient for clinical use, new, highly sensitive HBcrAg measurements should be applied to observe HBeAg-negative patients.

In conclusion, HBcrAg and M2BPGi are useful novel biomarkers for predicting HCC development in CHB. These markers would be useful for future medicine based on novel anti-HBV treatment targeting cccDNA and fibrolysis agents targeting liver fibrosis. Especially, because serum HBcrAg reflects the amount and replication activity of intrahepatic cccDNA, it will be generalized for the treatment responses and disease progression of CHB patients. Further international studies from many directions are required to increase the application of these assays for many aspects of CHB clinical practice worldwide.

## Figures and Tables

**Figure 1 ijms-21-00949-f001:**
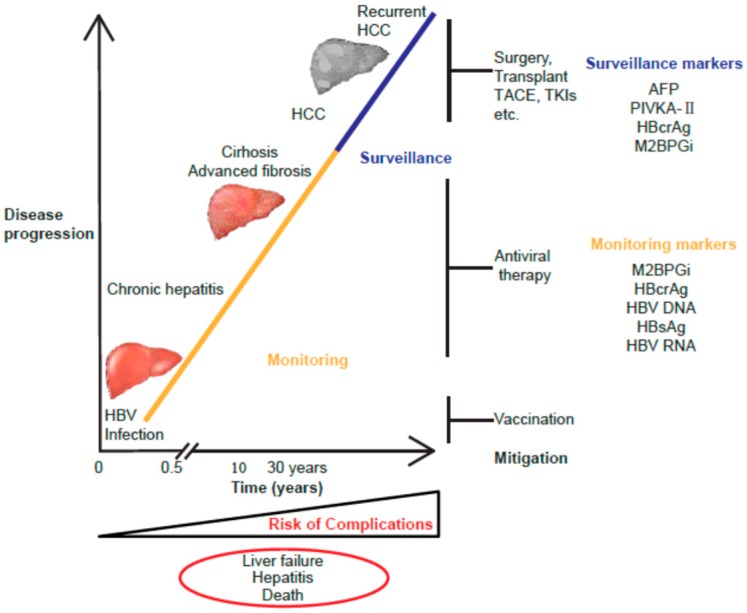
Chronic hepatitis B virus (HBV) infection related to liver disease progression. The schematic shows the clinical stages involved in the natural history of chronic hepatitis B (CHB). The novel serum biomarkers hepatitis B core-related antigen (HBcrAg) and Mac-2 binding protein glycosylation isomer (M2BPGi) provide valuable prognostic data for effective management of CHB. It is important to monitor the patients at high risk and to treat them in order to prevent liver complications, cirrhosis, and hepatocellular carcinoma (HCC) development. Abbreviations: HBV, hepatitis B virus; HCC, hepatocellular carcinoma; TACE, transcatheter arterial chemoembolization; TKIs, tyrosine kinase inhibitors; AFP, alpha fetoprotein; PIVKA-II, protein induced by vitamin K absence or antagonist-II; HBcrAg, hepatitis B core-related antigen; M2BPGi, Mac-2 binding protein glycan isomer; HBsAg, hepatitis B surface antigen.

**Figure 2 ijms-21-00949-f002:**
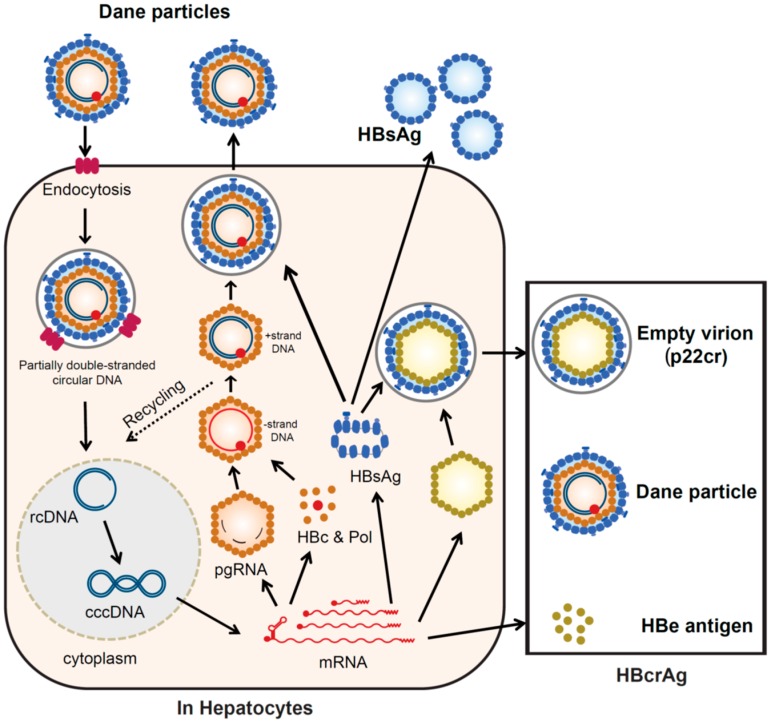
HBV replication cycle. The schematic depicts the steps involved in the HBV replication cycle. It shows the sources of various HBV-related molecules used routinely for diagnosis, clinical monitoring, and prognosis of HBV infection. These include HBcrAg (p22cr, HBeAg, hepatitis B core antigen (HBcAg), HBsAg, and HBV DNA, which can be measured in serum. Covalently closed circular (ccc)DNA is mainly measured from liver biopsy samples. Abbreviations: HBV, hepatitis B virus; HBcrAg, hepatitis B core-related antigen; p22cr, precore protein; HBeAg, hepatitis B envelope antigen; HBcAg, hepatitis B core antigen; HBsAg, hepatitis B surface antigen; cccDNA, covalently closed circular DNA; rcDNA, relaxed circular DNA, pgRNA, pregenomic RNA; HBc and Pol, hepatitis B core antigen and polymerase.

**Figure 3 ijms-21-00949-f003:**
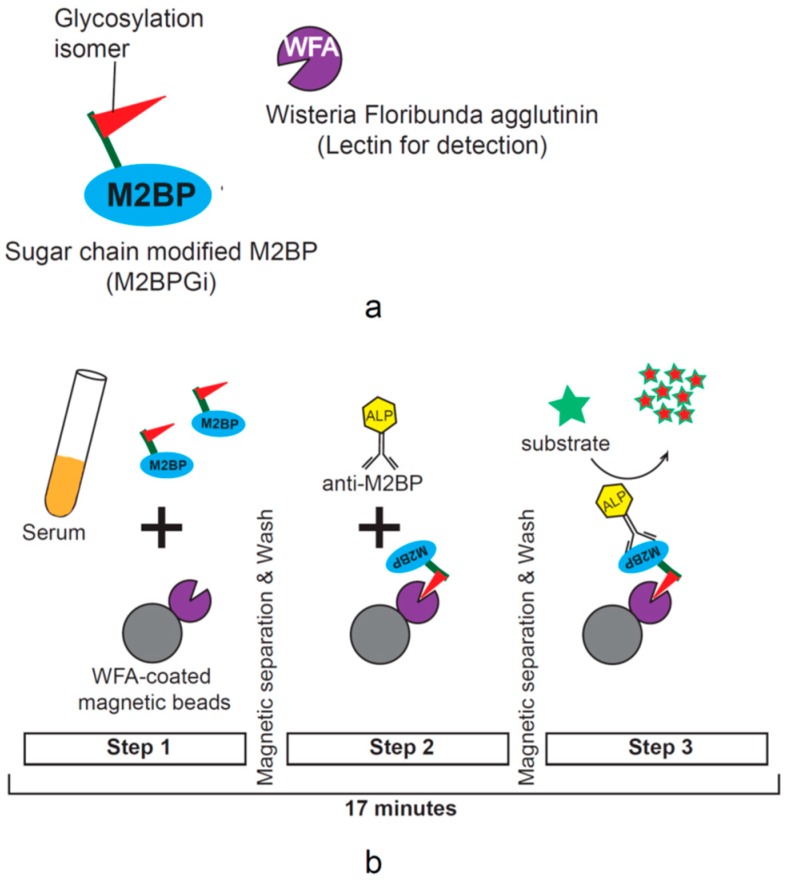
(**a**) Mac-2 binding protein (M2BP)–*Wisteria floribunda* agglutinin-positive (WFA) interaction. M2BPGi is a glycobiomarker of liver fibrosis with a unique fibrosis-related glycol-modification. The lectin WFA binds specifically to M2BPGi. This specific lectin–glycan interaction was exploited to develop a lectin-antibody sandwich immunoassay for the quantification of M2BPGi. (**b**) Automated serum M2BPGi quantification. The schematic depicts the main steps involved in M2BPGi quantification using the fully automated HISCL-2000i immunoanalyzer (Sysmex Co., Hyogo, Japan). Lectin immobilized on magnetic beads is mixed with diluted serum. ALP-labelled anti-M2BP monoclonal antibodies are then added after washing away unbound proteins. Chemiluminescent substrate and stop solution are added following a wash step to remove unbound antibodies. The automated assay takes only 17 minutes. Abbreviations: M2BPGi, Mac-2 binding protein glycan isomer; WFA, *Wisteria floribunda* agglutinin; ALP, alkaline phosphatase.

**Table 1 ijms-21-00949-t001:** Clinical applications of HBcrAg in CHB patients.

Category	Findings	HBcrAg level (log U/mL) and point	References
cccDNA activity	Lower amounts of intrahepatic cccDNA and lower cccDNA activity	<3 log U/mL	[44]
	Identification of inactive carriers with a high accuracy (any HBV genotype)	HBcrAg ≤ 3 log U/mL plus HBV DNA ≤ 2000 U/mL	[45]
HCC occurrence/recurrence	Incidence of HCC for treatment-naïve patients	>2.9 log U/mL during the follow-up period	[48]
	At high risk for HCC with intermediate viral load (HBV DNA 2000–19,999 U/mL)	≥4.0 log U/mL	[49]
	Incidence of HCC for treatment-experienced patients	>4.67 log U/mL at pre-treatment, >3.89 log U/mL at post-treatment	[50]
	HCC development during NA treatment	Detectable HBcrAg during NA treatment	[51]
	Long-term effect of NA treatment on HCC development	Higher serum levels of HBcrAg and BCP mutations were associated with HCC development, independent of NA therapy	[52]
	Evaluation of HCC occurrence	HBcrAg > 3.0 log U/mL and HBsAg > 3.0 log U/mL (cut-off values)	[53]
	HCC recurrence within 2 years	>4.8 log U/mL at the time of HCC diagnosis	[54]

Abbreviations: CHB, chronic hepatitis B; HBcrAg, hepatitis B core-related antigen; HBV, hepatitis B virus; cccDNA, covalently closed circular DNA; HCC, hepatocellular carcinoma; NA, nucleos(t)ide analog; BCP, basal core promoter; HBsAg, hepatitis B surface antigen.

**Table 2 ijms-21-00949-t002:** Clinical applications of M2BPGi in patients with chronic liver disease.

Category	Findings	M2BPGi level (C.O.I.) and point	References
Liver fibrosis	Significant fibrosis (≥F2) in CHB	≥1.06 C.O.I. (AUC 0.753)	[79]
Survival rate	Lower cumulative survival rate in patients with cirrhosis	≥5.0 C.O.I.	[78]
HCC development	A >15-fold-increased risk of HCC development in CHC patients after SVR	≥2.80 C.O.I.	[73]
	Higher risk of HCC development in CHB patients	≥1.8 C.O.I. for patients without cirrhosis (*p* < 0.001) ≥1.8 C.O.I. for patients with cirrhosis (*p* = 0.073)	[83]
	Low risk of HCC in HBeAg-negative patients	≤0.68 C.O.I. at baseline	[85]
	Risk for HCC development in CHB patients with cirrhosis treated with NAs	M2BPGi-based score* ≥ 652.5 at baseline	[86]
	Risk for HCC development in CHB patients treated with NAs	≥1.215 C.O.I. at 48 weeks	[87]
	HCC development in CHB patients	≥0.69 C.O.I.	[88]

*M2BPGi-based score, 8 × age (year) + 7 × baseline M2BPGi (C.O.I.) + 10 × body mass index (kg/m^2^). The score was calculable in 171 CHB patients with cirrhosis with a median of 652.5 (IQR, 581.3, 709.4) points. Abbreviations: M2BPGi, Mac-2 binding protein glycosylation isomer; C.O.I., cut-off index; CHB, chronic hepatitis B; AUC, area under the curve; HCC, hepatocellular carcinoma; CHC, chronic hepatitis C; SVR, sustained virological response; HBeAg, hepatitis B envelope antigen; NAs, nucleos(t)ide analogues; IQR, interquartile range.

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
