# Peer review of "Novel Biomarkers of Hepatitis B and Hepatocellular Carcinoma: Clinical Significance of HBcrAg and M2BPGi"

_ijms, 2020, doi:10.3390/ijms21030949_

Round 1

Reviewer 1 Report

Dear Authors

This is a very inderesting paper that point out the role and the significance of novel biomarkers in clinical assessment and follow up HBV patients, regarding risk of fibrosis and HCC development or recurrence . Maybe the use of these biomarkers,HBcrAg and M2BPGi, will be valuable at this time and in the near future, when newer  therapies will be apply to  HBV patients.

I believe this is an inderesting  paper that will contributes to new and novel clinical practice assessment and follow up HBV patients . I don't suggest any modification .

Author Response

Response to Reviewer 1 Comment

Reviewer 1

This is a very interesting paper that point out the role and the significance of novel biomarkers in clinical assessment and follow up HBV patients, regarding risk of fibrosis and HCC development or recurrence. Maybe the use of these biomarkers, HBcrAg and M2BPGi, will be valuable at this time and in the near future, when newer therapies will be apply to HBV patients.

I believe this is an interesting paper that will contributes to new and novel clinical practice assessment and follow up HBV patients. I don't suggest any modification.

Response: We thank you very much for your careful reading and for your comments.

Reviewer 2 Report

The review by Baudi et al. on the topic of biomarkers specifically on the clinical significance of HBcrAg and M2BPGi has been reviewed by me.

The review has been written to a high standard and displays the depth in which the authors have read through to write up this concise report.

The article itself is very readable and the authors have put forward their arguments on the significance of HBcrAg and its relevance as a marker for HCC.  However, as rightly pointed out there seems to be more work done using cohort in East Asia vs other parts of the world.

It would be interesting to see if the same would be observed in other large cohorts.

Similarly the glycosylated Mac2 binding protein is of interest and as shown by other authors shows  high correlation with HCC.

The article is well written and I would recommend its acceptance.

I however have a couple of minor points to raise.

1. In fig 1. which shows the replicative cycle of HBV could the authors also add in their figure the +ve stranded DNA and -ve stranded DNA during the replication of the virus in the cytoplasm.

2. Overall the figures look good, however could a better quality of image be used.

3. on page 15 line 446 is a typographical error developing should be developed.

Author Response

Response to Reviewer 2 Comments

Reviewer 2

The review by Baudi et al. on the topic of biomarkers specifically on the clinical significance of HBcrAg and M2BPGi has been reviewed by me.

The review has been written to a high standard and displays the depth in which the authors have read through to write up this concise report.

The article itself is very readable and the authors have put forward their arguments on the significance of HBcrAg and its relevance as a marker for HCC. However, as rightly pointed out there seems to be more work done using cohort in East Asia vs other parts of the world.

It would be interesting to see if the same would be observed in other large cohorts.

Similarly the glycosylated Mac2 binding protein is of interest and as shown by other authors shows high correlation with HCC.

The article is well written and I would recommend its acceptance.

I however have a couple of minor points to raise.

Response: We thank you very much for your careful reading and for your fruitful suggestions.

In fig 1. Which shows the replicative cycle of HBV could the authors also add in their figure the +ve stranded DNA and -ve stranded DNA during the replication of the virus in the cytoplasm.

Response: According to your comment, we modified Fig. 2. We added the negative stranded DNA and positive stranded DNA during HBV replication in the cytoplasm.

Overall the figures look good, however could a better quality of image be used.

Response: According to your comment, the figures with better quality were prepared. They will improve our manuscript.

on page 15 line 446 is a typographical error developing should be developed.

Response: As you pointed out, conventional hepatitis B core-related antigen (HBcrAg) has already been approved for clinical practice in Japan. However, new HBcrAg assays with approximately 10-fold higher sensitivity are now being developing, not developed. Based on this situation, we would like to leave the word “developing.”

We look forward to hearing from you regarding our submission. We would be happy to respond to any further questions and comments you might have.